# The Acceptability of a Tobacco Dependency Treatment for NHS Staff in the North East of England: A Mixed-Methods Study

**DOI:** 10.3390/ijerph22030352

**Published:** 2025-02-27

**Authors:** Caitlin Louise Thompson, Kerry Brennan-Tovey, Caitlin Robinson, Rachel McIlvenna, Eileen F. S. Kaner, Sheena E. Ramsay, Maria Raisa Jessica Aquino

**Affiliations:** 1NIHR Applied Research Collaboration North East North Cumbria, Population Health Sciences Institute, Faculty of Medical Sciences, Newcastle University, Newcastle-upon-Tyne NE1 7RU, UK; 2Gateshead Health NHS Foundation Trust, Gateshead NE9 6SX, UK; 3North East North Cumbria NHS Integrated Care Board, County Durham and Darlington NHS Foundation Trust, Darlington DL3 6HX, UK

**Keywords:** smoking cessation, healthcare services, tobacco dependency

## Abstract

Aims: High smoking rates and deprivation levels in the North East of England have led to an opportunity to pilot a tobacco dependency treatment offer for NHS (National Health Service) staff who smoke, to make a supported quit attempt. The direct and indirect benefits to staff, patients, and NHS organisations are well documented. This study aimed to evaluate service acceptability. Methods: The service included up to 12 weeks of free nicotine replacement therapy (NRT) and/or a refillable e-cigarette, motivational support, and premium access to the Smoke-Free app. The service evaluation used a mixed-methods design, combining the Theoretical Framework of Acceptability (TFA) questionnaire and semi-structured interviews with staff who had accessed the offer. The quantitative data were analysed using descriptive statistics and qualitative data via thematic analysis. Results: Sixty-eight survey responses reflected high acceptability and revealed four themes relating to the service familiarity and ease of access, suitability of the NRT/E-liquid ordering service, the vape kit, and behavioural support. Conclusions: The service was deemed highly acceptable, and service users’ experiences informed recommendations for improving future tobacco dependency services. This is the first known application of the TFA to an evaluation of a smoking cessation intervention, and it contributes to a broader body of research on reducing tobacco dependency.

## 1. What This Paper Adds

### 1.1. What Is Already Known on This Topic?

The North East of England is one of the most deprived regions in the country, contributing to higher than average national smoking prevalence rates;Encouraging National Health Service (NHS) staff to quit smoking can help promote smoke-free environments amongst staff and the wider community, contributing to the goals of the NHS Long Term Plan;A unique opportunity arose to support staff across an NHS Integrated Care Board spanning England’s North East and North Cumbria regions.

### 1.2. What This Study Adds

Limited research exists on the acceptability of staff smoking interventions within healthcare services, and none to date have used validated acceptability measures. This study provides insight into this gap by using mixed methods.

### 1.3. How This Study Might Affect Research, Practice, or Policy

By understanding the barriers and facilitators faced by NHS staff accessing stop smoking services, this study provides recommendations for improving future services to ensure that interventions remain acceptable and continue to achieve the desired outcomes.

## 2. Background

The UK’s National Health Service (NHS) is the fifth largest employer in the world, with NHS England employing 1.5 million staff in February 2024 [1]. While smoking rates amongst healthcare staff are typically lower than the general population, a substantial number of NHS staff are likely to smoke [1].

The Hiding in Plain Sight report [1] highlights the association between smoking, reduced life expectancy, and increased likelihood of hospital admission and premature death caused by smoking-related diseases among NHS staff who smoke. This report also captures that staff smoking costs the NHS approximately £206 million each year, comprising £101 million from sickness absence, up to £99 million from smoking breaks, and £6 million in sickness treatment costs [1]. Providing support for staff smoking cessation can help improve health outcomes and work productivity, as well as save the NHS from the avoidable costs of treating tobacco-related illnesses [1].

The North East North Cumbria (NENC) NHS Integrated Care Board (ICB) is one of the largest integrated care systems in the UK, employing 170,000 staff members. Across the NENC ICB, staff smoking contributes to 36,710 hospital admissions and 14,288 premature deaths in the region annually [2]. This costs the NHS approximately £2.8 million in productivity, social, healthcare, and fire costs across NENC ICB [2]. Addressing staff smoking remains a priority within the region, and doing so can help support the implementation of the NHS Long Term Plan by promoting smoke-free environments for both staff and patients [3].

In the NENC region, smoking prevalence rates remain higher than the England average (13.0% vs. 12.7%, respectively) [1]. The NENC region has one of the highest deprivation levels in the country [2]. Although rates are trending downwards, figures continue to mask inequalities, with higher rates of smoking amongst those with lower socioeconomic status, and smoking continues to be the leading driver of health inequalities [4]. As such, smoking prevalence remains high amongst routine and manual workers, with 21.5% of routine and manual workers being smokers across the NENC Integrated Care Board (ICB) [5]. Supporting staff across all work settings to quit smoking remains a priority to address health inequalities in the region.

Limited access to quit smoking support through work has been identified as a barrier to successful quit smoking attempts [6]. Under national pilots funded by NHS England, the Greater Manchester Integrated Care Partnership launched a tobacco treatment offer for Greater Manchester-based NHS employees. The offer included free access to the Smoke-Free app, 12 weeks of nicotine replacement therapy (NRT), and/or a refillable vape device for staff [7]. The results demonstrated reduced smoking rates, measured as 12-week abstinence rates, by 37–39% [7]. The highest quit rates were reported for NRT/vape use in combination with the app, compared to the app alone [7]. Although promising, these findings do not lend insight into the acceptability of the service from the users’ perspective, including the barriers to and enablers of engagement. Providing this evidence could help inform on the sustainability of implementing such a service to staff working in NHS settings.

In light of NICE guidance [8] and the priorities outlined in the NHS Long Term Plan [3], the NENC ICB implemented the NHS Staff Tobacco Dependency Offer (STDO) alongside the rollout of NHS England-funded tobacco dependency treatment services in acute inpatient, mental health inpatient, and maternity settings. The current evaluation aimed to address gaps in the evidence on the acceptability of tobacco dependency services for NHS staff by exploring the extent to which the NHS Staff Tobacco Dependency Offer (STDO) was acceptable for staff in the North East and North Cumbria (NENC) region and identify any barriers to/enablers of engagement that may affect successful quit rates. The study reported here—an expansion on a conference proceeding by Thompson et al. [9]—provides the first known insight into the acceptability of smoking cessation services delivered to healthcare staff using a validated theoretical framework.

## 3. Methods

### 3.1. The NHS Staff Tobacco Dependency Offer

The NENC STDO was piloted from December 2021 to September 2023, with 1972 people taking up the offer across the NENC region. For context, at the time of the evaluation (September 2023), the NENC ICB employed 81,117 staff [10]. The service, however, also included staff employed by subsidiary companies, which may not be reflected in NHS workforce statistics. This is worth noting, given the high proportion of routine and manual workers within this setting; therefore, the number of employed staff may not accurately reflect the number of staff eligible for the service.

The offer provided NHS staff who smoke with up to 12 weeks of free NRT products and/or refillable e-cigarettes, as well as access to free 12-week motivational support with a trained stop smoking advisor. Access varied across the region; for example, some areas had specialist services, such as a team of advisors who saw smokers through a 12-week quit attempt on site. In contrast, other areas commissioned services (i.e., pharmacies or GP practices) to deliver the intervention on their behalf. All staff were offered full premium access to the Smoke-Free app [11] for nine months, which offers remote, flexible support 24/7.

### 3.2. Design and Procedure

A mixed-methods evaluation was undertaken, consisting of quantitative cross-sectional survey work and qualitative semi-structured interviews. Participants were recruited by email and through the stop smoking service. Recruitment posters were shared through internal communications by members of the NENC ICB team. The survey took up to 15 min to complete (online and paper copies were available). At the end of the survey, participants were given an opportunity to leave a contact email address/phone number to participate in an interview. Participants were offered a £15 voucher following interview completion. Participants could also contact researchers to participate in an interview without completing the survey. Informed consent was obtained from all participants. Data collection occurred between July and November 2023. The interviews were audio recorded and transcribed verbatim (by an external transcription company). Following the interviews, participants were provided with a debrief form.

### 3.3. Participants

Convenience and snowball sampling techniques were used to recruit participants who met the following eligibility criteria: staff employed by NHS hospitals across the NENC region, including those employed on subsidiary contracts; those who self-identify as smoking tobacco products or as having previously smoked tobacco products; those currently engaging with the NENC STDO service or having engaged with but withdrawn from the NENC STDO service or having engaged with and completed the course of treatment provided by the NENC STDO service. Recruitment was informed by information power [12].

### 3.4. Survey

#### 3.4.1. Data Collection

Acceptability was measured using a survey instrument based on the Theoretical Framework of Acceptability (TFA) [13]. The TFA questionnaire is a validated, brief, and adaptable tool used to measure intervention acceptability across a range of healthcare settings [14]. The TFA consists of eight constructs: Affective Attitude, Burden, Ethicality, Effectiveness, Coherence, Self-Efficacy, Opportunity Costs, and General Acceptability. Participants responded to the survey on a Likert scale from 1 to 5 depending on the context of the question, e.g., strongly disagree to strongly agree [13].

#### 3.4.2. Data Analysis

The data were managed using IBM SPSS Statistics (Version 27). The questionnaires were analysed descriptively, using absolute and relative frequencies. Descriptive statistics (mean (*M*) and median (*Mdn*) along with standard deviation (*SD*), standard error (*SE*)), and 95% confidence intervals for each of the eight TFA constructs were obtained. Higher scores on TFA constructs are suggestive of higher acceptability [14]. Three of the eight constructs were reverse scored (Burden, Ethicality, and Costs).

### 3.5. Interviews

#### 3.5.1. Data Collection

Semi-structured, one-to-one qualitative interviews were used to gather the experiences and perspectives of the service users who had engaged with the NENC STDO. Interviews followed a topic guide, specifically designed to evaluate the current service, informed by the TFA [13]. Interviews lasted approximately 30–45 min and took place over Microsoft Teams or telephone, depending on participant preference.

#### 3.5.2. Data Analysis

The interview data were managed using NVivo Version 12.0 [15] and were analysed using thematic analysis [16]. Deductive coding built upon the eight TFA constructs alongside inductive coding (i.e., coding from the data) to identify the key themes relating to acceptability and user experience that might not otherwise fit into the TFA constructs. The authors recognise that their own experiences and perspectives may have influenced theme development, and to ensure rigour, two authors (CT and KBT) independently coded 10% of the transcripts. They agreed upon a shared codebook, which CT used to code the remaining transcripts.

## 4. Findings

### 4.1. Survey Findings

A total of 133 responses (110 online; 23 paper) were received. The removal of duplicates, incomplete, or invalid responses (*N* = 65) resulted in a final sample of *N* = 68. Of the 68, 64 respondents provided their age (*M* = 43.92, *Mdn* = 43.50, *SD* = 12.27). Table 1 provides an overview of the participant characteristics.

The survey results revealed that participants reported feeling ‘comfortable’ to ‘very comfortable’ in engaging with the STDO (Affective Attitude, see Table 2), (*M* = 3.877, *SD* = 1.552, and *Mdn* =5.00). Participants also found it easy to engage with the intervention (Burden) (*M* = 4.108, *SD* = 1.331, and *Mdn* = 5.00), and participants, on average, had ‘no opinion’ on whether the STDO had “ethical or moral consequences” (Ethicality) (*M* = 3.123, *SD* = 1.390, and *Mdn* = 3.00).

Furthermore, participants tended to ‘agree’ that the service aided a quit attempt (Effectiveness) (*M* = 4.462, *SD* = 0.801, and *Mdn* = 5.00). Staff also reported having a strong understanding of the package of treatment and support offered by the service (Coherence) (*M* = 4.477, *SD* = 0.783, and *Mdn* = 5.00), and generally felt confident engaging with the service (Self-Efficacy) (*M* = 4.477, *SD* = 0.763, and *Mdn* = 5.00). Responses also indicated that the NENC STDO did not interfere with existing priorities (Opportunity Costs) (*M* = 3.754, *SD* = 1.480, and *Mdn* = 4.00).

Finally, the survey revealed high overall acceptability (*M* = 4.585, *SD* = 0.715, and *Mdn* = 5.00) for the general acceptability of the service. For an overview of the descriptive statistics for each construct, see Table 3.

### 4.2. Interviews Findings

A total of 18 interviews were conducted. Participants’ ages ranged from 23 to 63 years (*M* = 40.67, *Mdn* = 37.5, and *SD* = 12.22). Table 4 details the interviewee demographic characteristics.

### 4.3. Themes from Qualitative Study

Four themes were identified relating to the familiarity and ease of service access, suitability of the NRT/E-liquid ordering service, suitability of the vape kit E-Liquid product, and suitability of the behavioural support offered by smoking cessation advisors. Each of the themes are discussed in turn below, with illustrative quotes provided. Quotes are presented with participant numbers, gender (M or F), and whether the staff worked in clinical or non-clinical roles to uphold anonymity.

#### 4.3.1. Familiarity with the Service and Ease of Access

Some participants noted that the service was well advertised, whilst others reported that more staff needed to be made aware of the service depending on whether they were office-based or patient-facing. For example, participants noted that the service was advertised well for computer-based access, e.g., using QR codes, emails, and online staff pages, which could limit the type of workers it reached. However, once participants were acquainted with the service, accessing was thought to be easy, particularly for those with dedicated support on site. Staff also reported convenience, as they could access it through work, which motivated engagement:

“No. It’s on our intranet… but obviously some people who we work with, say domestics and stuff like that, don’t really access the computers that often, and they might not know… I think sometimes, as well, having posters up and stuff like that… where people who haven’t got as much IT skills can still see it” (P005, F, Clinical);

“Yeah, so it was very easy to do. Because I think they’re here from two o’clock on a Thursday until four o’clock so you can just pop down whenever you’re free” (P008, F, Non-Clinical).

#### 4.3.2. Suitability of the NRT/E-Liquid Ordering Service

Participants reported general satisfaction with NRT and E-Liquid options, e.g., the amount offered from the initial point of access and throughout the duration of the 12 weeks. Participants were positive about being able to access their desired form of NRT. Furthermore, participants reported the general acceptability and speed of the service’s ordering and delivery service for E-Liquids:

“There was quite a good choice of liquids really…any flavour that you wanted in your milligrams of nicotine that you wanted” (P015, M, Clinical);

“So I’ve got in touch, and it was really efficient, I filled out some paperwork and that, and then I think it was within two days I got a vape that came through with a month’s liquid, which was great” (P005, F, Clinical).

#### 4.3.3. Suitability of the Vape Kit Offered

Participants who accessed the e-cigarette reflected on the quality of the vape kit offered. For example, some experienced the technical issues associated with their use. Many, however, complimented the vape kit as providing a promising starting point for their quit journey, with others reporting the vape kit as promoting a continued quit beyond the service. The free vape kit was also seen as a motivating factor for the participants to begin their quit journey:

“…the first vape that I got given wouldn’t charge so initially it was fine but then it wouldn’t charge… I ended up having to ring and they gave me some advice… And that got that going again and then I replaced it a third time. In those three months, I replaced the vape three times.” (P013, F, Clinical);

“I think it’s a good service and as I say I think for me I obviously got the kind of free equipment and juice to get started but from that, that’s just been the kind of, the start really of kind of kickstarting me into stopping and it has been successful for me”. (P003, M, Non-Clinical).

#### 4.3.4. Suitability of the Behavioural Support Offered (Smoking Cessation Advisors and App)

Many participants described good communication with smoking cessation advisors, commenting that the regular contact helped to facilitate a successful quit. Some participants suggested that they could have benefitted from a stronger rapport being built, possibly with advisors having greater knowledge and experience (e.g., on the effects of vaping). Also, some participants reported wanting further behavioural support from the service.

There were mixed views regarding the modalities of accessing the service. Participants found telephone contact with smoking cessation advisors less acceptable than other modalities, e.g., face to face. However, some reported accessing in-person, drop-in smoking cessation advisor appointments as challenging (e.g., preferring a private location for appointments). Participants suggested methods of improving accessibility, for example, signposting to the peer support available, e.g., through the Smoke-Free app or creating peer networks to connect with other people accessing the service.

Regarding the Smoke-Free app, some participants reported being able to utilise tips offered on the app to aid their quit, whilst other comments suggested that it may not be suitable for supporting a quit attempt alone, working best in conjunction with other support, e.g., behavioural and/or NRT:

“I think that I anticipated more. I mean obviously you’re not going to do too much because people are busy but I thought there would have been more… even at nearly 40 you still like a little bit of praise every now and then” (P003, M, Non-Clinical);

“Like a cessation support group, with other smokers going through, we can go and talk about their experiences…Then from a support group people create like WhatsApp chats or teams chats or whatever…” (P001, F, Clinical).

## 5. Discussion

The findings revealed that overall, participants found the service to be acceptable. Descriptive statistics from the survey indicate that the service was deemed acceptable across seven of eight validated constructs using the TFA [14]. The survey findings contribute to a limited body of evidence using the TFA for evaluating smoking cessation services delivered in hospital settings for healthcare staff. The findings also report on which domains can be targeted to ensure the acceptability of future iterations of the intervention. The qualitative findings reflect potential barriers to service engagement and provide insight into the service users’ experiences of the product and support offered. Barriers such as service adverts were raised, with patient-facing staff possibly benefiting more from word-of-mouth/poster advertisement. Furthermore, the quality of the vape kit was reported on, alongside the importance of solid rapport building with advisors. Enablers of the service, such as the ease of NRT ordering, access to a range of choices, and the fact that the service was free and available through work, were reported as key strengths.

Previous studies have shown that smoking cessation services, which offer a combination of vape/NRT and behavioural support from an app, can help motivate successful quits and facilitate the move towards a smoke-free NHS workforce [6], in line with the goals set out by the NHS Long Term Plan [3]. Furthermore, a pharmacy-supported e-cigarette programme in the North West found that service users greatly appreciated the free device given, with financial savings playing a pivotal role in the motivation to engage in a quit smoking attempt [18]. Similarly, a scoping review, a systematic review, with a meta-analysis of workforce tobacco dependency services found that workplace interventions are effective methods of motivating a quit smoking attempt, recognising financial costs as key barriers to implementation [19,20].

Previous studies have also demonstrated that strong motivational counselling has been found to improve long term goal attainment in workplace stop smoking services [18]. Having non-stigmatising, positive behavioural support has also been found to reduce potential feelings of shame/stigma felt by staff seeking support from embedded workplace smoking cessation services [21].

It is understood, therefore, that free, accessible services that offer a combination of NRT/e-cigarette and behavioural support can successfully motivate quit smoking attempts. Overcoming barriers to obtaining stop smoking support, such as embedding stop smoking services in the workplace, as recommended by NICE guidelines [8], is also evidenced as effective for service adherence. Less, however, is understood about the acceptability of embedded stop smoking services within NHS workplaces. This study, therefore, adds to the current understanding by providing the first known application of the TFA to evaluate the acceptability of an intervention package for smoking cessation embedded within NHS work settings.

The findings from this evaluation can be used to tailor future smoking cessation services for healthcare staff more generally. For example, the findings highlight the need for services to be widely promoted using a combination of word-of-mouth, posters, and email/internet-based adverts, with consideration for patient-facing workers with limited email/computer access. This is particularly important given that smoking patterns are highly socioeconomically patterned, with those working in non-clinical settings, i.e., routine and manual workers, demonstrating higher smoking rates [1]. Further work on service uptake, e.g., staff profession, to fully assess the reach of the service, is required, which was beyond the scope of the current evaluation.

In support of the findings from Greater Manchester [6], qualitative accounts suggested that the Smoke-Free app is insufficient to support quitting smoking alone; rather, it is best used in conjunction with NRT and/or behavioural support. This is in line with the current NICE guidelines, which recommend a combination of NRT and behavioural support (individual and group) in smoking cessation services for an increased likelihood of positive clinical outcomes [8]. The interview findings also revealed a need for strong communication and rapport with advisors providing behavioural support. Participants additionally provided recommendations to strengthen the behavioural support offered by the service, for example, through establishing network connections with other service users, to provide and receive peer support. Participants also noted different preferences for contacting smoking cessation advisors, i.e., face to face/telephone, which could have facilitated rapport.

The following practice and policy recommendations for improving the acceptability of future stop smoking services for healthcare staff have been developed, which could be applicable to wider smoking cessation services: service promotion should be made widely accessible across different settings and include a variety of formats; to maintain service user engagement, formal and informal support should be available (e.g., on-site advisors, peer networks, and bookable appointments); to support service delivery, smoking cessation advisors should have regular training to ensure that the knowledge and information given is based on contemporary evidence; finally, to maintain service quality, opportunities for providing feedback should be made available and used to inform service improvement. The findings also have potential clinical implications, as improving such services for healthcare staff can help to facilitate smoke-free environments across staff and patients, thus improving health outcomes across both groups.

### Limitations of This Study

The study has some limitations. Firstly, there were challenges in recruiting participants for both the survey and the interviews due to the timing of the evaluation, i.e., there was a reduced number of staff accessing the service at the point of evaluation, leading to a smaller pool of potential participants. Challenges were overcome using different recruitment channels (i.e., email, posters, and word-of-mouth) and working closely with key contacts within services to recruit participants. It is also worth noting the drop-out rate from the survey sample, as although 133 survey responses were received, only 65 eligible responses were included in the analysis, accounting for incomplete responses and/or duplicates. The drop-out rate may reflect the feasibility of survey completion. It may also reflect the views of a limited pool of participants, i.e., those motivated to complete or those who had a positive experience of the service.

Furthermore, using the TFA presented challenges, for example, applying constructs to the evaluated service (e.g., the Ethicality construct). For example, the original TFA survey was trialled on the delivery of a COVID-19 vaccine [14]. The Ethicality construct, i.e., an intervention’s fit with an individual’s values [13], may be limited due to the moral and ethical differences in and consequences of vaccination acceptability compared to that of a smoking cessation service. Future research is recommended to improve the application of TFA constructs for smoking cessation and other similar interventions to ensure the accessible and applicable wording of survey questions. Additionally, it was beyond the study’s remit to validate the service outcomes, i.e., using quit data either as self-reported quit or as validated quit (CO monitoring).

The authors recognise the potential influence of their prior knowledge and experience on theme development. Thus, the authors engaged in reflexivity across the research process, for example, through considering the relationships between the researchers and participants. CT, who conducted interviews, had no prior relationship with the participants and is a non-smoker. In addition, the team, who are from varied backgrounds (psychology and physiotherapy), had regular meetings to reflect on data collection, analysis, and interpretation, to ensure that broad perspectives were captured and interrogate gaps in the data.

## 6. Conclusions

The evaluation demonstrates that the NENC STDO was deemed acceptable by NHS staff. Service users found the NRT/vape products and behavioural support to be accessible, with many service users wanting the service to be sustained, possibly for extended periods, with greater amounts of behavioural support being offered. Further research on services across different healthcare settings, and potentially further analysis of service uptake across work professions, is needed. The current study provides the first known application of the validated TFA framework to a smoking cessation intervention.

## Figures and Tables

**Table 1 ijerph-22-00352-t001:** An overview of survey sample demographics (*N* = 68).

Demographic		*N* (%)
Gender	Male	19 (27.94)
	Female	49 (72.06)
Ethnicity	White—British	66 (97.06)
	White—Other	1 (1.47)
	Black or Black—British/African	1 (1.47)
Local Authority/Service Accessed	Newcastle Council (provided by Change Grow Live)	4 (5.88)
	North Tyneside Council	2 (2.94)
	Hartlepool Council	1 (1.47)
	Stockton-on-Tees Council	2 (2.94)
	Sunderland Council	2 (2.94)
	Northumberland Council	3 (4.41)
	Gateshead Council (including QEF outpatient pharmacy)	2 (2.94)
	Tees, Esk and Wear Valley (TEWV) NHS Foundation Trust	22 (32.35)
	North Cumbria (provided by Gateshead Health Staff Team or NHS Smoke-Free App)	11 (16.18)
	Durham County Council (provided by ABL Health) *	4 (5.88)
	Smoke Free Staff Team—Gateshead Health	4 (5.88)
	Do Not Know	7 (10.29)
	South Tees Stop Smoking Service	0 (0.00)
	South Tyneside Council	0 (0.00)
	Durham County Council (provided by ABL Health) and TEWV (selected two)	4 (4.88)
Work Setting	Allied Healthcare Professional	7 (10.29)
	Nursing and Midwifery	13 (19.12)
	Community Services	2 (2.94)
	Clinical Support Staff/Healthcare Assistant	17 (25.00)
	Admin and Clerical	14 (20.59)
	Porting and Estates	5 (7.35)
	Domestic Services and Catering	4 (5.88)
	Corporate Services	1 (1.47)
	Medical Professional	2 (2.94)
	Other	1 (1.47)
	No response	2 (2.94)

* Note: Staff working or living in Darlington would have accessed Durham County Council’s service provided by ABL Health.

**Table 2 ijerph-22-00352-t002:** TFA constructs, definitions, and whether the construct is reverse scored.

TFA Construct	Definition	Reverse Scored
Affective Attitude	How comfortable individuals felt engaging in the NENC STDO.	No
Burden	How effortful individuals perceived the NENC STDO to be.	Yes
Ethicality	Reflects the extent to which engaging in the NENC STDO had moral or ethical consequences.	Yes
Effectiveness	The extent to which the intervention was perceived to have achieved its objective, i.e., aid in a quit attempt.	No
Coherence	The extent to which participants understood how the intervention worked.	No
Self-Efficacy	How confident individuals felt about engaging in the NENC STDO.	No
Opportunity Costs	The extent to which engagement in the NENC STDO interfered with other priorities.	Yes
General Acceptability	Overall acceptability of the service.	No

**Table 3 ijerph-22-00352-t003:** Summary of descriptive measures of each TFA construct from survey results.

TFA Construct	Mean (SD)	Median	Standard Error	95% CI
Affective Attitude	3.877 (1.552)	5.00	0.196	3.486–4.268
Burden	4.108 (1.331)	5.00	0.161	3.785–4.430
Ethicality	3.123 (1.390)	3.00	0.172	2.780–3.467
Effectiveness	4.462 (0.801)	5.00	0.101	4.260–4.663
Coherence	4.477 (0.783)	5.00	0.082	4.312–4.641
Self-Efficacy	4.477 (0.763)	5.00	0.096	4.285–4.668
Opportunity Costs	3.754 (1.480)	4.00	0.185	3.385–4.123
General Acceptability	4.585 (0.715)	5.00	0.090	4.405–4.765

**Table 4 ijerph-22-00352-t004:** Interview sample demographics (*N* = 18).

Demographic		*N* (%)
Gender	Male	7 (38.89)
	Female	11 (61.11)
Ethnicity	White—British	16 (88.89)
	Black or Black—British/African	2 (11.11)
Local Authority Accessed	North Tyneside Council	2 (11.11)
	Northumberland Council	1 (5.56)
	Tees, Esk and Wear Valley (TEWV) NHS Foundation Trust	3 (16.67)
	Smoke Free Staff Team—Gateshead Health	4 (22.22)
	Unknown	8 (44.44)
Work Setting	Nursing and Midwifery	5 (27.78)
	Clinical Support Staff/Healthcare Assistant	6 (33.33)
	Admin and Clerical	5 (27.78)
	Domestic Services and Catering	1 (5.56)
	Corporate Services	1 (5.56)
Trust Employed at	Northumbria Healthcare	3 (16.67)
	North Cumbria Integrated Care	5 (27.78)
	Cumbria, Northumberland, Tyne and Wear	2 (11.11)
	Tees, Esk, Wear and Valley	6 (33.33)
	Newcastle Hospitals	2 (11.11)
UK Educational Level *	Level 1	3 (16.67)
	Level 2	0 (0.00)
	Level 3	5 (27.78)
	Level 4	0 (0.00)
	Level 5	3 (16.67)
	Level 6	5 (27.78)
	Level 7	2 (11.11)

* Examples of UK Education Levels as per Gov.uk website [17]; Entry Level: Entry Level Awards, Level 1: GCSE grades 3, 2, 1 or grades D, E, F, G, Level 2: GCSE grades 9, 8, 7, 6, 5, 4 or grades A*, A, B, C, Level 3: A-Levels, Level 4: certificate of higher education (CertHE), Level 5: foundation degree, Level 6: degree apprenticeship or degree with honours (e.g., BA or BSc), Level 7: master’s degree (MSc or MA).

## Data Availability

Anonymous survey data (as an SPSS file) will be shared upon reasonable request by the corresponding author. The interview data cannot be shared publicly due to transcripts containing potentially identifying material and, therefore, cannot be shared to protect the anonymity and privacy of participants.

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
