# Peer review of "The Acceptability of a Tobacco Dependency Treatment for NHS Staff in the North East of England: A Mixed-Methods Study"

_ijerph, 2025, doi:10.3390/ijerph22030352_

Round 1

Reviewer 1 Report

Comments and Suggestions for Authors

Dear Authors,

You are working in an interesting and important area.  Below are comments to consider as you continue work on the manuscript:

--Most commentary in the Background section addresses costs to the NHS.  Perhaps there should at least be mention of costs of individuals and families in terms of suffering, premature death, etc.

--Long term or long-term?  Check all uses for consistency.

--Section 3.1 -- "This is worth noting, given the portion of routine and 103
manual workers within this subgroup of staff." -- not entirely clear

--Section 3.1 -- "within a Trust." -- What does this mean?

--Section 3.3 -- "staff employed or a subsidy of one of the NENC NHS Trusts" -- may shed some light on question just above, but this statement isn't clear; perhaps a word or words is/are missing

--Section 3.4.2 -- "bespoke topic guide" -- What does this mean?

--Section 3.5.1 -- "for each of the seven TFA constructs" -- Why 7 (section 3.4.1 indicates 8 constructs)? (Section 3.5.2 also refers to  7 constructs.)

--Section 3.5.2 -- "The authors recognise that their own experiences and perspectives may have influenced theme development" -- need to address in limitations

--Section 5 -- "Similarly, both a scoping review, a systematic review, and a meta-analysis of workforce tobacco dependency services" -- both?

Again, your work is interesting, and I enjoyed reading your article.  Best wishes as you revise the work.

Reviewer 2 Report

Comments and Suggestions for Authors

I believe that the results of the study on how NHS staff accept interventions for tobacco dependence provide valuable information and knowledge to understand the acceptability of smoking cessation interventions among staff in similar fields in other regions beyond the specific area. However, there are quite a few points that need to be improved and supplemented in relation to the writing of the manuscript. Here are somethings I would like you to improve it:

1. Although most people in professional circles will know what the abbreviation NHS is, for those readers who don't, the full name should be stated when it is first presented in the abstract.

2. There is no need to present basic statistics in the abstract.

3. I understand that this journal is not like other journals that require "What this paper adds" to be presented before the background. It is very important to explain what this research contributes or the research gap in the introduction. Incorporate it into the background or introduction.

4. Aren't surveys and interviews "Procedures" rather than "Materials"?

5. It is not acceptable to have the same subtitles in different numbers in a paper or formal report (e.g. 3.4.1. Survey = 3.5.1. Survey).

6. There is a need to further clarify what descriptive outcome measures can contribute.

7. The clinical implications need to be further elucidated.

Reviewer 3 Report

Comments and Suggestions for Authors

The article evaluates the acceptability of a tobacco dependency treatment program implemented for NHS staff in the North East of England, highlighting the significant public health implications of smoking within this demographic. The study was motivated by the high smoking rates and levels of deprivation in the region, which contribute to increased health inequalities and healthcare costs associated with smoking-related illnesses. The research utilized a mixed-methods approach, combining quantitative surveys and qualitative interviews to assess the acceptability of the NHS Staff Tobacco Dependency Offer (STDO). The service provided up to 12 weeks of free Nicotine Replacement Therapy (NRT) and/or refillable e-cigarettes, along with motivational support and access to the Smoke-Free App. The findings indicated a high level of acceptability among participants, with an overall mean score of 4.59 on the acceptability scale, suggesting that the service was well-received by NHS staff. Qualitative data revealed four key themes regarding user experiences: familiarity with the service, ease of access, suitability of the NRT and e-liquid ordering service, and the effectiveness of behavioral support provided by smoking cessation advisors (Page 66). Participants expressed satisfaction with the accessibility of the service, noting that it was well-advertised through various channels, although some suggested that more could be done to reach non-clinical staff who may not have regular access to digital communications. The study also identified barriers to engagement, such as the need for stronger rapport between service users and advisors, and the importance of providing comprehensive behavioral support alongside pharmacological interventions. Participants highlighted the necessity of ongoing support and communication to enhance their quit attempts, indicating that while the service was effective, there is room for improvement in the delivery of behavioral support. In conclusion, the article contributes valuable insights into the acceptability of smoking cessation services within NHS settings, emphasizing the need for tailored approaches that consider the unique challenges faced by healthcare staff. The findings suggest that while the STDO is a step in the right direction, further enhancements in service delivery and support mechanisms are essential for maximizing its impact on smoking cessation among NHS employees.

There is some little misspelling mistakes int he article like:

  1. "behavioural" is spelled with a "u" in British English, which is correct throughout the document. However, if American English is preferred, it should be changed to "behavioral" (Page 22).

  2. "e-liquid" is consistently spelled as "E-Liquid" in some instances, which should be standardized. For example, it appears as "E-Liquid" on Page 66 but is also referred to as "e-liquid" on Page 33.

  3. "advertisement" is spelled correctly, but in some contexts, it could be simplified to "advert" for brevity (Page 88).
